# Physical Activity for Children and Youth with Physical Disabilities: A Case Study on Implementation in the Municipality Setting

**DOI:** 10.3390/ijerph19105791

**Published:** 2022-05-10

**Authors:** Charlotte Boslev Præst, Thomas Skovgaard

**Affiliations:** 1Active Living, Institute of Sports Science and Clinical Biomechanics, University of Southern Denmark, 5230 Odense, Denmark; tskovgaard@health.sdu.dk; 2Research and Implementation Centre for Human Movement and Learning, University of Southern Denmark, 5230 Odense, Denmark

**Keywords:** physical activity, children and youth, physical disabilities, municipal practices, implementation

## Abstract

Children and youth with disabilities participate less in physical activity (PA) than their peers. This qualitative multiple-case study aimed to assess how municipal practices support PA implementation for these children and youth. A total of 23 interviews were conducted, which covered participants from different departments and professional positions in two municipalities. Local policy documents were included. A thematic analysis was performed, which was based on Winter’s integrated implementation model and Gittell’s theory on relational coordination. The study findings indicate how PA implementation for children and youth with disabilities is a complex challenge that involves several departments and agents. The study demonstrates that having an explicit policy that focuses on parasport can positively influence organisational and interorganisational behaviour, and that coordinating consultants seem to play an essential role in PA implementation. However, there is room for improvement at the municipal level in order to promote a better overall performance in terms of the support for PA participation of children and youth with disabilities.

## 1. Introduction

It is well known that participation in physical activity (PA) benefits physical, psychological, and social health. For children and youth with disabilities, PA participation holds the potential to positively impact quality of life, self-esteem, and identity-feeling, and it may also be the path to increased participation in other life situations [1,2,3,4,5,6].

PA refers to movement in many different forms and arenas, such as active play, active transportation, sport, and rehabilitation at home, in school, in therapy, or as a leisure-time activity. As stated in the Convention on the Rights of Persons with Disabilities (UNCRPD) by the United Nations (UN), participation in PA is a human right for children and youth living with disabilities, and they must experience equal opportunities for this [7]. Nonetheless, compared to the population at large, this group participates less in PA, being 16–62% less likely to meet PA guidelines [5,8,9,10].

Numerous studies have identified the barriers to and the facilitator of PA participation among people with disabilities, with all of them concluding that multiple factors play a role. The factors include: the attitude and perceived benefits; self-perception; time and energy; the feeling of belonging; the degree of support from family, professionals, and peers; the capacity of relevant organisations to support population groups with disabilities; the available opportunities; and possibilities for active transportation [5,11,12,13,14,15,16]. To improve the opportunity for PA participation, a multilevel approach is required that is aimed at the individual level as well as at the social, community, and policy levels [5,15,16,17]. To meet this complexity, different studies have incorporated such multilevel factors into conceptual models. A recent comprehensive systematic review by Martin Ginis et al. [5,15] organised these factors into five levels to highlight the different determinants that have an impact on the PA participation among people with disabilities. Similarly, Shields and Synnot [16] conducted their study of barriers and facilitators, and they divided their suggestions for strategic interventions into different levels (individual, social, and policy). While it is evident that promoting PA for children and youth with disabilities is a complex task—which requires a multidisciplinary and holistic approach—not much is known about how this complex task is handled and realised in real-life settings [5].

Therefore, this study aims to assess how the practices in local settings support the implementation of PA possibilities for children and youth with physical disabilities, including how such practices and implementation processes are affected by a number of facilitators and barriers. The study is a multiple-case study that focuses on two Danish municipalities, and it starts with a specification of the Danish municipal structure, followed by a description of the theoretical framework that is applied in the study, which is the integrated implementation model by Søren Winter and Jody Gittell’s theory on relational coordination.

### 1.1. The Danish Political and Administrative Structure

Denmark is one of the more decentralised countries in Europe, with the municipalities being relatively autonomous authorities with: elected councils that control the local executive structure; the power to make local political decisions, which are regulated by rather broad national legislation; and an independent source of taxation. In general, the municipalities in Denmark have rather large populations in comparison to other European countries, with an average size of 56,735 citizens, and a number of municipalities with populations greater than 100,000 citizens [18]. This can be contrasted to the total population in Denmark, which is currently 5.8 million people. In short, the Danish structure makes it so that municipalities are major forces that are free to organise the local administrative design and to decide what functions should be allocated to which departments. A core responsibility of Danish municipalities is to provide welfare services to its citizens, such as the promotion of health and rehabilitation; primary school, including special education; culture; sports, etc.

Since 2006, it has been compulsory for all municipalities to have a disability council that provides guidance to politicians, authorities, and organisations on how to improve conditions for people with disabilities [19]. Furthermore, it is recommended that a disability policy that contains an action plan be developed in order to ensure that the guidelines in the UNCRPD are met. According to Danish law, municipalities are obligated to offer free counselling and services to children and youth with disabilities and their families (Section 11 in the Act on Social Services 2018) [20].

With the municipalities being responsible for managing physical and social affairs for children and youth with disabilities, this level of government makes an interesting case to study in the attempt to describe and understand the practices that affect the local implementation of PA for children and youth with physical disabilities.

### 1.2. Theoretical Framework: Policy Implementation and (Inter) Organisational Behaviour

In this section, the theoretical starting points of the study will be presented.

#### 1.2.1. Søren Winter’s Integrated Implementation Model

The study applies Søren Winter’s integrated implementation model, which is an explanatory framework that presents the key factors that influence the implementation outputs and outcomes [21]. Winter is, among others, inspired by Michael Lipsky’s [22] theory on street-level bureaucrats (SLBs) as important policymakers, and he tailors this by emphasising how managers and organisational behaviour also are key factors in implementation processes, and, in the end, in the municipal performance, which is understood as the outputs of municipal activities in relation to, for instance, support for the PA participation of children and youth with disabilities. Thus, the performance is seen in a broad sense, and it contains specific services, counselling, grants, etc.

##### The Street-Level Bureaucrats’ Role in Policy Implementation

SLBs are fieldworkers who interact directly with citizens when implementing public policies [21]. The theory is based on the underlying assumption that the behaviour of SLBs is driven by: (a) the SLBs’ incentives to act (the motivations and interest towards the target group and the subject area in question; e.g., PA for children and youth with physical disabilities), and (b) their capacities to act (time, resources, competencies, and knowledge) [21]. SLBs generally work in situations that are characterized by multiple demands and limited resources. This conflict is typically managed by applying a set of coping mechanisms [21,23]:Decrease demands for SLB performance by limiting information about services, which makes access difficult, and which imposes a variety of other psychological costs on the client;Ration services by giving higher priority to one type of service over others. This often occurs when other objectives are more clearly expressed (e.g., formalized academic goals compared to broadly worded ambitions) that are related to PA for all students in primary and lower secondary school;Standardise and routinise SLB work by dividing clients into categories instead of delivering individualised treatment or counselling.

Coping mechanisms can be practical in making tasks manageable, but, at the same time, they can make them dysfunctional through the systematic distortion and hindering of policy implementation and goal achievement, and, possibly, through a lack of receptiveness to citizen needs and inputs [21].

##### Role of Management

Management plays a key role in the implementation of policiesamong other things, by prioritising and coordinating the actions of SLBs, amongst other things. Particularly, first- and middle-level management are important in supporting the performances of SLBs [21,24]. More generally, management supports implementation processes by turning organisational visions and strategies into concrete actions, and by ensuring that a broad variety of interests and needs interact in the realization of workable solutions and developments. Management is often centred around creating effective connections and prioritising between a number of framework conditions, such as laws, regulations, or core organisational values and relationships that are linked to, for instance, collaborations with key staff teams, user groups, and other managerial levels.

#### 1.2.2. Organisational and Interorganisational Behaviour

The need for a multilevel approach with regard to PA for children and youth with disabilities requires both organisational and interorganisational collaborations. In other words, for the implementation of possibilities for PA participation to succeed, collaborations and coordination across municipal departments and broader sections of local communities are needed.

On the basis of studies on the performance of the objectives that demand collaboration, Jody Gittell [25] presents her theory on relational coordination. This theory emphasises two important dimensions for strong performances and interorganisational collaboration: communication and relations. Communication should be frequent, timely, precise, and problem-solving, and good relations are characterised by shared objectives, knowledge sharing, and mutual respect.

Winter’s implementation model is designed for the analysis of specified policy implementation processes (most times comprising a longer sequence of events, decisions, and actions). However, in this study, the model is used as a general framework to describe how municipal agents continuously work for the implemention of PA for specific target groups, as is required by national and international law. This adjusted use is deemed as an applicable minor alteration of the original approach. Gittell’s theory on relational coordination is applied to capture the impact of (inter) organisational behaviours. Interestingly, Gittell has applied her theoretical framework to study a specific Danish municipality setting that is working to incorporate relational coordination into the implementation of initiatives on healthy aging, and it was found usable in this complex system, with collaborations across many departments [26]. Studies that apply Winter’s theoretical model have mainly focused on the practices of social workers, both with vulnerable children and youth [23], as well as with other target groups that are different from the ones in this study (refugees, employment policies, etc.). Moreover, for such reasons, this study is rather innovative.

The specific aim is to answer the following research questions (RQ):(1)How do local policies support the implementation of PA for children and youth with physical disabilities?(2)How do Danish municipalities—at the organisational and interorganisational levels—work with for the implementation of PA possibilities for children and youth with physical disabilities?(3)How does the behaviour of municipal agents and the coordination across departments affect the performance in relation to ensuring PA possibilities for children and youth with physical disabilities?

## 2. Materials and Methods

### 2.1. Study Design

The study is a qualitative multiple-case study that was conducted in two Danish municipalities: Municipality 1 (M1) and Municipality 2 (M2). The evidence from multiple and perhaps even related, but different, cases is often considered to be more compelling because it increases both the empirical and the analytical strength and provides a more solid foundation for deduction [27]. A high level of generalisability is not the primary goal in a case-study design. However, the use of multiple cases, an explicit and thorough application of theory, and transparency in the research process and choices increases the potential recognisability and transferability of a study such as this [27,28]. A clear strength is the use of multiple data sources, which reaches a deeper understanding through data triangulation. The data sources that are used in this study are interviews with municipal agents and local policy documents.

### 2.2. Setting and Participants

The two municipalities are of similar sizes, with around 100,000 citizens each. They are geographically located in different parts of Denmark: one in the capital area, the other in the region of Southern Denmark. One municipality was chosen on the basis of its own declaration as “The Parasport Capital”, and it can thus be classified as a crucial case [27]. The other municipality was chosen for comparison.

In both cases, the local disability councils acted as the first contact. The chairpersons were solicited, after which, in both municipalities, contact was made with the representatives who are engaged in parasport. From here, the participants were included by using chain sampling in order to ensure that they fit into the study aim. Participants were selected on the basis of their professional (SLBs and managers, referring to Winter’s implementation model) and organisational positions in order to cover the relevant departments within the municipalities. Managers were contacted and interviewed, whereupon they suggested and forwarded the information of one or more employees (SLBs) to take part, all with years of employment and with tasks concerning PA and/or the target group. However, from the first contact, in both municipalities, it was clear that it was also essential to include a third group: agents who having more coordinating functions within the municipality (subsequently referred to as “municipal consultants”). These municipal consultants are, similar to other municipal agents, employed by local authorities. Typically, the term “consultant” is used to point out that such staffers handle job functions that are related to organisational and/or activity coordination, development, etc.

The participants all agreed to take part in the study, and they were provided with the opportunity to receive information on the results. Overviews of the participants are presented in Table 1 and Table 2.

### 2.3. Data Collection

#### 2.3.1. Interviews

A total of 24 interviews were conducted between November 2020 and February 2021, which lasted between 36 and 53 min. One interview was not included in the final material, since this participant only worked with adults. Two of the interviews within M1 were not recorded, since the participants called directly in response to an email sent by the first author, and they only wanted to participate in an interview at that specific time. Notes were taken from the two interviews. Another interview was planned as a preliminary interview to obtain access to M1 with three participants from the local parasport club. Detailed notes were taken both during and following the interview since it became clear that these participants played a larger part in the organisation than was first anticipated (Figure 1). All other interviews followed a semistructured interview guide that was based on the theoretical framework. This was modified to fit the specific agent (SLB, manager, or consultant). An example of the interview guide can be found in Appendix A. Interviews were conducted as online video calls, and the audio was recorded.

#### 2.3.2. Policy Documents

Local policy documents served as a secondary source of empirical material, and they supported and expanded the primary source (interviews) by increasing the understanding of the points that were made by the participants, and they provided insight into the organisational and decision-making structures. The documents were accessed via the municipal websites and/or by information through interviews. Documents were systematically examined and further included in the process if they were directly relevant to the study aim.

### 2.4. Analytical Strategy

The 20 recorded interviews were transcribed, and the names and places were anonymised. The empirical material was then analysed by using thematic content analysis, as an iterative process [29]. The coding was initially kept open, which was followed by grouping in themes by primarily taking a deductive approach. An example of the coding can be found in Appendix B. The data analysis was conducted with qualitative-data-analysis software (NVivo 12). The local policy documents were read through, and the relevant phrases were highlighted, which was followed by a selection of the information to be presented in the tables. All of the interviews were performed in Danish. The translation of the supporting citations into English was performed after analysing all of the data (interviews and documents).

### 2.5. Ethics

The research was conducted in accordance with the Declaration of Helsinki. The research protocol was sent to The Regional Committees on Health Research Ethics for Southern Denmark, and it was exempted from further ethical approval (reference number: 20202000-191). Furthermore, the project was submitted to and approved by The Research and Innovation Organisation of the University of Southern Denmark (notification number: 11.207). Prior to the data collection, participants were informed about the project aims, the content, and the participants’ legal rights, both in writing and orally. All data are stored and treated in accordance with Danish law for data protection. All of the participants were anonymised, and they are only referred to by profession and department. Similarly, the municipalities are referred to as M1 and M2. To secure anonymisation, the documents are referred to as “internal documents”.

## 3. Results

The findings are structured in line with Figure 2, and they respond to the research questions (RQ) and to the theoretical framework.

### 3.1. Organisation and Local Policies

This section responds to RQ1: “How do local policies support the implementation of PA for children and youth with physical disabilities?”, and to RQ2: “How do Danish municipalities—at the organisational and interorganisational levels—work for the implementation of PA possibilities for children and youth with physical disabilities?”.

#### 3.1.1. Municipality 1 (M1)—A Brief Description of the Organisation of Local Authorities

M1 is organised into three main administrative-policy areas: Children and Youth; City, Culture, and Environment; and Social Affairs, Health, and Labour. All three are subdivided into several departments. The majority of the tasks that are relevant for children and youth with disabilities are located in the departments under Children and Youth: The School Department, the Family Department, and the Department of Pedagogical and Psychological Consultation (from here on, referred to as PPR, which is the general reference in Denmark).

The School Department is responsible for the local school system, which includes both the “ordinary” schools and special-education services. The director of the School Department participates in the local educational committee, where local policies are discussed, and major developments are initiated. Additionally, the director participates in a visitation committee, which is responsible for granting personal assistance at schools if needed.

The PPR consists of different professions, with physio- and occupational therapists being of special interest for this study. The PPR has a dual function: (1) a consultative function in the school setting, and (2) a treatment function, which performs therapy/training with children who have been referred to the Children Training Clinic (embedded within the PPR) from caseworkers in the Family Department.

The Family Department consists primarily of caseworkers. Their work is highly regulated by the Act on Social Services, which determines how the department can and must grant support to the child and family. Via the Act on Social Services, the municipality is required to cover additional expenses that are due to the disability in relation to transportation, leisure-time activities, and the parents’ losses in earnings, and it must offer free counselling and rehabilitation [20]. These three departments, which are located under Children and Youth, collaborate regularly, and primarily on a case-by-case basis.

The Culture and Leisure Department, which is located under City, Culture, and Environment, likewise deals with PA opportunities for children and youth. However, the municipality differs from most by outsourcing the employment of sports consultants to the general local sports organisation and to the local parasport club.

Thus, the department does not have any direct contact with children and youth with disabilities; however, as the special consultant explains it:

The department does not contribute directly to children and youth with disabilities’ participation in PA, but uses the local Disability Policy to determine whether projects suggested by the local sports clubs can be approved.(Special consultant, M1. Notes from phone conversation.)

The employment of a sports consultant in the local parasport club is, for instance, grounded in the Disability Policy.

The Department of Social Affairs, Health, and Labour primarily targets adults. However, specialised functions, such as tailored devices and customised aids, such as wheelchairs, protheses, etc., are handled by this department, and also when providing such services to younger citizens.

#### 3.1.2. Municipality 1 (M1): The Influence and Usage of Policy Documents

Within the municipality, several local policy documents exist on PA and children and youth with disabilities. These policies are presented in Table 3. In accordance with the UNCRPD, all the policies highlight the equal right to participate in society, in general, and in PA, specifically [7]. Other returning focuses across the different policies are to take a holistic approach to every child, and the need for collaboration and the coordination of efforts across departments. In the “Coherent Child Policy”, it is specified that: “A good knowledge of each other helps create clarity in relation to opportunities for action, as well as coordination of interdisciplinary efforts when needed” (internal document, M1). However, when talking to the local parasport club, frustration is expressed with regard to what is seen as a lack of knowledge of their existence from exactly the department that that policy targets:

Funnily enough they do not know we exist. I mean, those who can actually send them [the target group: Children and youth with disabilities] in our direction, they do not know we exist./…/The Family Department is equally surprised every time we contact them. When there is a new manager, we are forgotten. That is how we experience it anyway. (Sports consultant, the local Parasport Club, M1.)

The Disability Policy is particularly relevant. The policy creates a basis for the yearly “Action Plan”, with specific initiatives. The pertinent initiatives are illustrated in Table 3. It is specified what department is responsible for each initiative: Initiative 1: The Family Department, and Initiatives 6, 7, and 8: The Department of Culture and Leisure. This makes the policy more tangible. Notably, the Culture and Leisure Department does not consider itself to be a direct contributor to the PA participation of children and youth with disabilities. At the same time, the department is responsible for several initiatives in the “Action Plan”. Thus, it is actually the department, to a large extent, that contributes to this area. This view is supported by the local parasport club:

She [the special consultant from Culture & Leisure] is our lifeline. I talk to her maybe four times a week. (Chairperson, the local Parasport Club, M1)

Their relation is characterized by having a shared objective (initiative from the ‘Action Plan’), where they are mutually dependent on each other for adequate performance.

It appears that the local policies form a solid basis for the implementation of PA. However, the knowledge and the usage of these policies varies across departments, and between managers and SLBs. Overall, managers are aware of the policies, but they consider them to be something that is more value-based:

We have the Children & Youth Policy, and it is the bar to meet for all of us—that all children have the right to a good child life. (Manager, the Family Department, M1)

On the other hand, the SLBs did not seem to know the different policies when asked about them:

The one you just showed me [the Disability Policy], I did not know it. (Physiotherapist, PPR ‘ordinary’ schools, M1)

Strikingly, those working within the school system are not very aware of the policies. At the special school, neither the manager nor the teaching assistant knew any of the local policies regarding PA. However, they both agreed that PA is well integrated into the school day. Similarly, the pedagogical PE consultant expresses:

I do not remember them [the policies]. And if it states anything about children with special needs or children with physical or mental disabilities, I actually do not remember. (Pedagogical PE consultant, School Department, M1)

#### 3.1.3. Municipality 2 (M2): “The Parasport Capital”—A Brief Description of the Organisation of Local Authorities

Similarly, M2 is divided into different main areas—three of them being of direct relevance to this context: Children and Youth; Culture and Health; and Welfare. In particular the first two are critical. The Welfare area is mostly concerned with adults; however, as with M1, the granting of, for instance, customised aids, is located here. Where the two municipalities differ, however, is that, in M2, the casework on special devices (e.g., wheelchairs, walkers, special bikes) is located under Children and Youth in the Department of Family and Handicap, and it is thus closer to other services for the target group. This arrangement, according to the therapist that is responsible for this casework, fosters good collaboration:

It provides good opportunities for competent feedback and discussions with the other caseworkers, to provide a holistic approach, in relation to the children and families. (Physiotherapeutic caseworker, Family & Handicap, M2)

Family and Handicap serves as the authority that performs casework on the basis of the Act on Social Services regarding support for the families and the coverage of additional expenses (as in M1). In M2, the PPR has an authority function that is additional to the counselling function at schools. They are responsible for allocating therapy/training services on the basis of paragraphs in the Act on Social Services. In summary, the casework is distributed between the different subdepartments within Family and Prevention (see also Table 2 for an overview).

Unlike M1, the special school in M2 has therapists that are employed to provide counselling and to participate in the daily routines at the school. Weekly training sessions are incorporated into the schedule. This employment at the school fosters collaboration between professions and a better implementation of PA.

What is exceptional for M2 is the political focus on parasport. In addition to the mandatory disability council, M2 has also had, since 2016, a parasport council, which the members refer to as “The Parasport Capital”. The council has a separate “Action Plan” with a parasport consultant, who is employed in Culture and Health and who is responsible for the hands-on implementation of the “Action Plan”.

#### 3.1.4. Municipality 2 (M2): “The Parasport Capital”—The Influence and Usage of Policy Documents

In Table 4, the selected objectives within this “Action Plan” are illustrated. Special attention is placed on increasing the awareness of parasport across departments, and on collaborations with schools, local clubs, and the national parasports federation: Parasport Denmark. Additionally, the municipality aims to host national and international (para)sports events, which they use strategically to place focus on parasport. One example is a current theme on “good transitions” between daycare and schools, which was inspired by the 2022 visit of the world´s largest multiple-stage bicycle race, the Tour de France, in Denmark:

The theme this year is called ‘The wheels are turning’. No matter what wheels you have, and how you move around, we can all participate in PA by some type of wheels./…/We chose this theme to include departments from Culture & Health. (Special consultant Children & Youth, M2)

Table 4 illustrates the local documents that are related to PA and the target group. The different policies focus on the creation synergies within and across departments, by taking a holistic approach in every individual situation, and on the creation of equal health opportunities for all.

Several participants mentioned how the political focus on parasport also makes it easier to implement practices that support PA for children and youth with disabilities:

That we have a clear vision to be the parasport capital forces us to move forward. It provides the possibilities to work with the area. An endorsement to spend energy on it, but also to go to other departments and say: ‘we actually have a political ambition, this is something all of us work on’. That is definitely a strength. (Manager, Sport, Event & Community, M2)

An example of this focus across departments is found in the Department of Sport, Event, and Community, where an established collaboration with Parasport Denmark has been expanded to include initiatives at schools, where experts run activities that are funded by the department. Additionally, the parasport consultant, who is employed in this department, has established collaborations between local sport clubs and the special school. For instance, the local golf club, in coordination with the school staff, established an 8-week course for some of the students, with great success.

Interorganisational networks on parasport exist across departments. However, these appear to be limited to the Department of Sport, Event, and Community and the School Department. Regular meetings take place across these departments. The Department of Family and Prevention, where the caseworkers for children and youth with disabilities are located, does not participate.

### 3.2. Practices within and across Departments (M1 + M2)

This section responds to RQ3: “How does the behaviour of municipal agents and the coordination across departments affect the performance with regard to ensuring possibilities for PA for children and youth with physical disabilities?”. The section is divided into the classifications from Winter’s integrated implementation model.

#### 3.2.1. Organisational and Interorganisational Behaviour—The Consultants

In both municipalities, the consultants (acting as coordinators) from different departments play essential roles in the implementation of policies and in securing interorganisational collaborations, although they are more visible in M2 than in M1.

In M1, the pedagogical PE consultant does not consider herself to have an influence on PA for the target group, and she does not point to any real issues with the few single-integrated children (children with disabilities that go to school with children without disabilities) within the “ordinary” schools. Nonetheless, a newly established collaboration between herself and the local parasport club (initiated by the parasport club) makes use of her position to secure knowledge on where the single-integrated children are (which is information that the parasport club does not have access to). The recruitment of this specific group of children is one of the initiatives in the “Action Plan” of the “Disability Policy”. In this collaboration, the focus is on recruitment for leisure activities, as well as on the establishment of a collaboration with regard to PE lessons at the “ordinary” schools to increase the inclusion of all students.

In M2, several consultants play important roles in the interorganisational collaboration processes and in the implementation of PA. The parasport consultant collaborates with several departments within the municipality and with the local sports clubs. He sees these collaborations as essential to the implementation of parasport:

It is difficult in my position, employed in Culture & Health, working with parasport but not having any direct interaction with the individual citizen./…/It is very much about being good in the communication flow and making your colleagues from other departments aware of the opportunities that exists. (Parasport consultant, M2)

The collaborations between the parasport consultant, the pedagogical consultant, and the chief consultant (Table 2) are especially important for the implementation of parasport within schools through the use of the networks of other schools within the different departments to create awareness of the opportunities.

However, in both municipalities, the interorganisational work on creating PA possibilities is hampered when the departments do not seem to share common objectives on this and/or related matters, or when the collaboration is not particularly important for the performance (i.e., there is a lack of interdependence). Contrary to this, collaborations within departments that directly depend on the services they offer (e.g., PPR, Family Department, and schools) seem more integrated. This is especially the case in M2, where principles that are specifically from relational coordination are being incorporated through internally organised courses and working groups.

#### 3.2.2. Management Behaviour

In M1, a focus on PA implementation for children and youth with disabilities is not apparent at the managerial levels. This can be contrasted to the “Local Health Policy”, which states that “Health promotion and prevention need to be prioritised by management, with distribution of resources in all departments” (Internal document, M1). Every department focuses on their individual tasks, with PA not being a particularly important focus. The only department that has PA as a main focus (Culture and Leisure) outsources the responsibility to the local parasport club and it is, thus, only directly involved to some degree. At the special school, however, PA is a priority. For instance, the school arranges and finances equine therapy, and it conducts a weekly athletics event together with the local parasport club. The school manager expressed the need for more resources to implement PA during the school day. Contrary to this, the director of the School Department states that the special school is already capable of dealing with this. Within the “ordinary” schools, the director believes the issue only concerns a small group of children that can be catered to at each individual school when required.

In contrast, PA for children and youth with disabilities is more of a shared focus in M2. Managers across departments support the implementation initiatives within this area, and they take explicit responsibility:

Any good interdisciplinary collaboration requires management support. And within the parasport area we have strong support, built over years and now well-established. (Manager, Sport, Event & Community, M2)

This view is, to some extent, shared by the other managers in M2, which prioritises manager networks across the departments on parasport. The manager of Family and Handicap does not, however, seem to take part in these networks. Furthermore, she does not consider the department to be very relevant for PA participation:

It is really not something we focus on in our daily work. (Manager, Family & Handicap, M2)

Apparently, these manager networks mainly work within the “operational” departments (e.g., school, leisure), and they do not incorporate departments that are responsible for cross-sectorial casework. It is, however, a focus point of Culture and Health to connect with other municipal departments to promote PA among children and youth with disabilities:

We are lucky to not be bound by so many restrictions as many of the other departments, which we try to utilise to be part of the solution. (Manager, Sport, Event & Community, M2)

For example, the department has made a platform with all the possible sports activities, and they try to put parasport on the agenda for the caseworkers from the disability area. The manager from the PPR is aware of these initiatives on the promotion of parasport but expresses that there are limited possibilities for action because of resource restrictions.

In both municipalities, the immediate responses from the managers within the departments that are responsible for different casework, and for the overall counselling of this particular group of children and families, that they do not view their departments as playing an important role in PA implementation. When asked directly, however, they do all agree that they contribute to some extent:

I think that where we have a focus is in relation to how we can compensate the families, so the children have the possibility to participate in some sport, if that is what they want. (Manager, Family Department, M1)

In both municipalities, they agree that covering extra costs in relation to special devices, such as racerunners (a three-wheeled running bike with body support), etc., is a way to support PA participation. However, they also state that PA is a very small focus area in their dealings with the children and their families.

#### 3.2.3. SLB Behaviour

Two overall types of SLBs emerge from the data analysis: those who do not seem to find it problematic to implement PA into their practice, and those who find it challenging or do not feel responsible for it. In many ways, SLBs are affected by their experiences of the incentives and capacities within the field, as described in Winter’s theory [21].

Overall, the SLBs from M2 expressed an interest in and a sense of responsibility for the promotion of PA among children and youth with disabilities—which has a positive influence on their behaviour—with all of them expressing a focus towards PA counselling as part of their job. Even the caseworkers whose managers expressed very limited focus on PA expressed that they were attentive towards the area. The same caseworkers, however, described a capacity challenge, which seems to affect behaviour, as well as the use of coping strategies, which result in a barrier to the support for PA participation. This was present for the caseworkers in both municipalities:

Our job is to a large degree to manage these paragraphs [from the Act on Social Services], with the options being what lies within these. It kind of sets the framework. (Caseworker, Family Department, M1)

Factors such as time, resources, and knowledge similarly limit the feeling of capacity. An attempt to manage these factors has been indicated to trigger the following coping mechanisms:Rationing services: Arguing that other services are of higher priority to explain why counselling on PA possibilities is given less of a priority;Standardising work: Fitting the target group into specific categories with regard to which sport activities can be supported and/or informed about. As the caseworker in M2 expressed:
We kind of have to put them into boxes, because you cannot go to both wheelchair hockey, football, basket, wheelchair everything.(Caseworker, Family & Handicap, M2)Decreasing demands: Both by limiting user information and by imposing psychological costs on the target group; for instance, by using the fatigue of the child or the time/resources of the parents as an explanation for the individual not participating in PA.

Several of these coping strategies play a role in multiple departments. Within the “ordinary” schools, the lack of knowledge and time is also expressed as a challenge. SLBs from the PPR in both municipalities express the same capacity challenges with limited resources and knowledge, but they have the motivation to increase their own and others’ insights and coordination:

I think we should focus more on how we can, to a larger degree, establish a dialogue and collaboration with the local organisations to utilise their connections and knowledge on possibilities. It is a lot of work for us to provide individual guidance in every case. (Physiotherapist, PPR, ‘ordinary’ schools, M1)

The perspectives that were put forward by the teaching assistants from the special schools in both municipalities constitute a particularly strong example as to how the attitude of the individual SLB can influence the professional behaviour towards the implementation of PA. When talking to the teaching assistant from M2 about incorporating PA into the school, even though she is not educated within the field, she stated:

I think I am maybe a bit like Pippi Longstocking, even if there is something I have not tried before, I can probably do it. And then I seek guidance with the ones who know more, or I use the internet. (Teaching assistant, special school, M2)

Similarly, in M1, the teaching assistant explained how her own interest and attitude towards PA impacts her behaviour at work:

I move a lot myself—bike, swim, and run—which I have tried to incorporate in my work the last few years./…/When you show that you think it is awesome to move, it transfers to the children. (Teaching assistant, special school, M1)

## 4. Discussion

This study confirms that municipal-based PA implementation for children and youth with disabilities is a complex, multifactorial challenge that involves several departments. Interestingly, the study indicates how a municipality with an explicit political focus on parasport—underscored by the appellation “The Parasport Capital”—can positively influence both organisational- and interorganisational behaviour, as was demonstrated by almost all of the departments that showed an awareness of this particular focus and that expressed interest in working to achieve the objectives that are related to this. Furthermore, the study indicates how consultants play an essential role in the PA implementation for children and youth with disabilities by enhancing knowledge-sharing and coordination. However, the organisational structures of the municipalities challenge the end performances.

In the following, the findings are discussed in relation to the theoretical framework: the components from the integrated implementation model and the relational coordination. This forms a basis for an elaboration on how municipalities can take future steps towards implementing possibilities for PA participation for children and youth with disabilities.

### 4.1. Policies with Action Plans—A Starting Point for Increasing the Implementation of PA Possibilities across Multiple Municipal Levels

Both municipalities have several policy documents that, to some degree, focus on PA and on children and youth with disabilities. How well these policies are implemented and how well they support PA possibilities for these children and youth differ between the two municipalities. As the recently published “global perspective” article by Martin Ginis et al. [5] emphasises, policies must go further than just mentioning the need. Specific action plans on how to ensure the implementation of PA participation for children and youth with disabilities that target the different multidimensional levels that are involved are crucial. As just described, the municipality with a pronounced political focus on parasport (M2) appears to have a stronger focus on the implementation of PA across departments, where the majority of managers and all of the SLBs show positive behaviours towards this ambition. This is in line with Winter’s theory on the important organisational factors for implementation, where manager support combined with solid competences and incentives for SLBs promotes solid implementation [21].

### 4.2. A Coordinating Function, Working across Departments, Is Key to High Performance

This focus across most departments in M2 is strongly facilitated by the parasport consultant who is employed under Culture and Health and who works primarily on the implementation of initiatives from the parasport council’s “Action Plan” across departments and local sports clubs. This coordinating function is one of the strategies that has been shown by Gittell [25] to increase the performance of specific objectives by improving the relational coordination. The coordinating role increases the relational coordination by facilitating usable communication (timely and problem-solving) and relations across the involved parties through the implementation of shared knowledge, mutual respect for each other’s functions, and, perhaps most importantly, shared objectives [25]. Through the job function, the parasport consultant works towards strengthening the capacities of the involved SLBs within the municipality and the agents from the local sports clubs by providing resources, further training, and specific activities, which thereby supports better conditions for positive behaviours towards PA implementation [21]. This takes place very much through collaborations with colleagues from other departments in the municipality. By contrast, in M1, the department that is in charge of many of the initiatives on PA from the “Action Plan” of the “Disability Policy” (the Department of Culture and Leisure) outsources the consultancy function to the local parasport club (outside the municipality setting). This arrangement complicates implementation processes and creates unclear communication paths, decreased knowledge-sharing, and marked challenges with regard to reaching the children and youth with disabilities who attend “ordinary” schools, especially. However, the newly established collaboration in M1 with the pedagogical PE consultant from the School Department has stimulated a shared focus on reaching these children. This organisational development, once again, indicates how the coordinating functions within a municipality can increase collaboration and communication, and can thereby influence the performances with regard to PA implementation.

Previous research has pointed to quality collaborations between local governments, community partners, and schools as a possible strategy to remove, or to at least diminish, the barriers for PA among children and youth with disabilities [15,16]. In many ways, both municipalities are working towards such an ambition, with M2, at the moment, being much further down the line, which is very much due to the organisation with and around the designated parasport consultant.

### 4.3. Identifying Shared Objectives and Collaboration across Different Municipal Levels—Room for Improvement

There is, most definitely, room for improvement when it comes to collaboration and the identification of shared objectives with regard to PA implementation for children and youth with disabilities, and especially in two areas within both municipalities: the “ordinary” school area, and the departments that work with casework. Both of these areas have been identified by previous research as challenged when it comes to the establishment of durable collaborations with regard to PA implementation [15,16]. At the same time, they are both crucial, given their direct contact with the children and their families. This is especially the case in relation to the provision of information on the possibilities in the local community, the opportunities for support, and by enhancing the personal attitudes of the child and the family towards PA [30].

### 4.4. Individual SLB Behaviour Influences Municipal Performance

This study supports Winter’s theory on SLB-behaviour-influencing implementation [21,23,31]. SLBs who have positive attitudes towards PA for children and youth with disabilities and, at the same time, rate themselves as competent, incentivised, and as having reasonable conditions for professional actions, appear more likely to focus on PA implementation in their work, whereas the SLBs who experience a lack of time, resources, and knowledge tend to use coping strategies to explain why PA is not a significant focus within their department (especially caseworkers).

### 4.5. Limitations and Further Research

With a complex task such as PA implementation, which demands a multilevel approach, the SLB behaviour is not the sole key factor in the overall performance. This seem to be one limitation in the application of Winter’s theory to this field, since the theory has primarily been used in more specific areas, with fewer departments involved, and with a focus on the implementation of one specific policy [21,23,31]. This study, however, compensates for this by incorporating the dimensions from Gittell’s theory on relational coordination, which captures the highlighted complexity and the multilevel challenges.

An additional limitation to this study is the exclusive focus on the practices that are performed by municipal agents. Obviously, it would have been of value to include the input and the assessments from the target group (including their parents), which would have added an important dimension with regard to how the municipal performance is actually experienced and processed by the group that is, in the end, to profit from this. Further research should consider looking into this dimension in order to supplement the findings on the municipal performance.

### 4.6. Recommendations for Practice

On the basis of the findings of this study, three recommendations for practice, particularly, can be put forward: (1) municipalities should formulate clear and interconnected policies with regard to why and how to implement PA for children and youth with disabilities; (2) municipalities should ensure sufficient and continued managerial engagement to support the SLB performance; and (3) municipalities should increase their focus on interorganisational collaborations; for instance, by employing agents with an explicit coordinating function.

## 5. Conclusions

When comparing municipalities, M2, with an explicit parasport focus, seems to have a larger degree of relational coordination and overall focus across departments. Perhaps this is due to the employment of a parasport consultant, which promotes interdepartmental collaboration and coordination with the sports clubs in the local community.

The findings from this study also indicate that noticeable political backing from the highest municipal level seems to support the performance in relation to the implementation of PA possibilities for children and youth living with disabilities.

In both municipalities, however, “ordinary” schools, which reach the single-integrated children, and the departments that work with casework, need to further their efforts if the aim is to contribute to building shared objectives and productive coordination in this area.

Hopefully, the findings from this study can inspire and support various stakeholders to strengthen their efforts to install high-performance collaboration across departments in order to establish quality, sustainable, and diversified possibilities for the PA participation of children and youth with disabilities.

Further research on how municipal practice can be developed so that children and youth with physical disabilities will experience opportunities to participate in PA is essential. Research that applies Winter’s integrated implementation model in co-designed and/or explorative studies (e.g., action research that involves municipal agents and the target group in the identification and implementation of specific actions) would be interesting and would provide useful knowledge for future implementation. Gittell’s theory on relational coordination constitutes a tested and theory-driven basis for collaborations across municipal departments and wider stakeholder groups.

## Figures and Tables

**Figure 1 ijerph-19-05791-f001:**
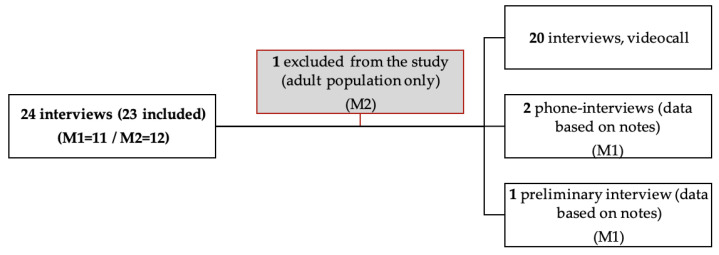
Flowchart, interviews.

**Figure 2 ijerph-19-05791-f002:**
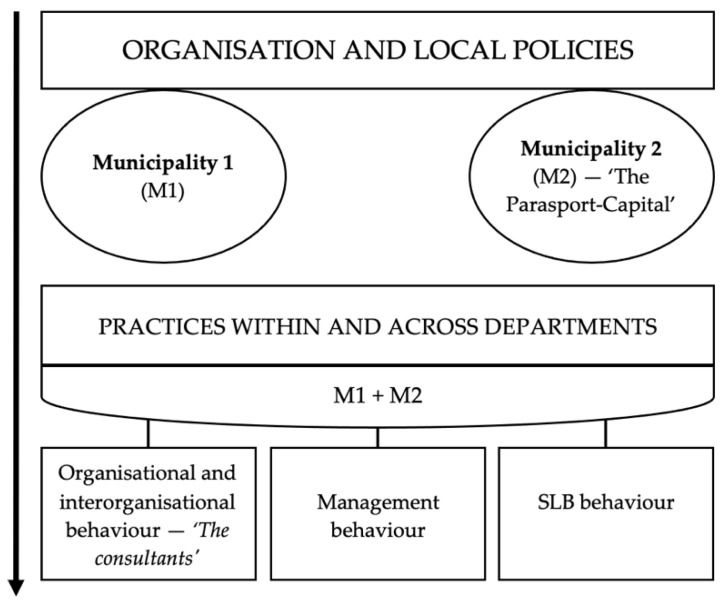
Structure of study findings.

**Table 1 ijerph-19-05791-t001:** Participants, Municipality 1 (M1).

Area	Department	Participants
**Children and Youth**	The School Department	Managers: Director Manager, special school
		SLBs: Teaching assistant, special school
		Consultants: Pedagogical PE consultant
	The Family Department, Section for Special Needs	Managers: Manager
		SLBs: Caseworker
	The Department of Pedagogical and Psychological Consultation (PPR)	SLBs: Physiotherapist, special school Physiotherapist, “ordinary” school
**City, Culture, and** **Environment**	The Department of Culture and Leisure	Special consultant
**Local Parasport Club**		Chairperson Sports consultant Volunteer
**Local Sports** **Organisation**		Sports consultant

**Table 2 ijerph-19-05791-t002:** Participants, Municipality 2 (M2).

Area	Department	Participants	
**Children and Youth**	The Children and Youth Staff Department	Consultants: Chief consultant	
	The School Department	Managers: Manager, special school + the children and youth training centre (located at the school) SLBs: Teaching assistant, special school Physiotherapist, special school + the children and youth training centre Consultants: Pedagogical consultant	
	The Department of Family and Prevention	Family and Handicap	Managers: Manager SLBs: Caseworker Physiotherapeutic caseworker
		Interdisciplinary Centre for Children and Youth (PPR)	Managers: Manager SLBs: Physiotherapist
**Culture and Health**	The Department of Sport, Event, and Community	Managers: Top manager Consultants: Parasport consultant	

**Table 3 ijerph-19-05791-t003:** Policy documents, Municipality 1 (M1).

Policy Documents, M1	Topics
**Disability Policy, 2019–2022**	Equal opportunity to participate in sports (including transportation, physical access, and information).
	An individual holistic approach that is coordinated across departments and the local community.
**Action Plan for the** **Disability Policy, 2020**	**Initiative 1:** A communication group for parents of children with disabilities. **Initiative 6**: Initiatives within the local parasport club: (1) funding for recruitment of schoolchildren with disabilities, and (2) swimming for children with cerebral palsy. **Initiative 7:** An open sport facility, facilitating participation for all. **Initiative 8:** Exercise for all—educate volunteers in the inclusion of people with disabilities.
**Health Policy, 2019–2022**	Reducing social inequalities in health. More children and youth should participate in PA.
	Cross-sectional collaborations, including support for local sports organisations.
**Coherent Child Policy, 2016**	Children and youth with disabilities should have the same opportunities to participate in society as their peers.
	A high priority on leisure activities for children with physical disabilities.
	A holistic and interdisciplinary approach across departments.
**Sport and Movement Policy,** **2019–2022**	All citizens should experience possibilities for PA.

**Table 4 ijerph-19-05791-t004:** Policy documents, Municipality 2 (M2).

Policy Documents, M2	Topics
**Disability Policy, 2019–2027**	Inclusion is a joint responsibility: the individual and family must feel adequately guided across departments.
	Equal opportunities for participation in the community, and to live healthy and active lives. The wish to be leading within parasport.
**Action Plan (Parasport Council),** **2020–2021**	Support local sports clubs to become capable of including citizens with disabilities by: (1) focusing on schools’ collaborations with local clubs and Parasport Denmark; (2) support the local clubs by strengthening competences; and (3) create awareness of the local parasport fund and application opportunities.
	The promotion of interdisciplinary collaborations within the municipality.
**Children and Youth Policy (disability),** **2013**	Facilitate self-esteem, health, and well-being—this is the start of growth.
	Children and youth with disabilities should experience the same opportunities to participate in leisure-time activities.
**Strategy for Family and Prevention** **2017–2020: “We want more—together”**	The strengthening of health promotion and interdisciplinary collaborations to create a coherent experience for families.
**Health Policy, 2017–2024**	All children and youth should have access to movement and sports activities. The involvement of the entire municipality in the achievement of health.
**Culture, Sports, and Leisure Policy:** **“The best place to live”, 2019–2022**	The municipality focuses on parasport. The sports environment should be accessible and open to all.

## Data Availability

The data are not publicly available because of legal and privacy issues. The data used in the research article are not available to anyone other than the authors in order to meet the requirements for anonymisation. Both authors have access to the original material.

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
