# Peer review of "Physical Activity for Children and Youth with Physical Disabilities: A Case Study on Implementation in the Municipality Setting"

_ijerph, 2022, doi:10.3390/ijerph19105791_

Round 1

Reviewer 1 Report

  1. Is there any language issue about translation in data collection? How do you deal with this issue?
  2. In 2.3.1, “All other interviews followed a semi-structured interview guide based on the theoretical framework.” It seems that a figure about the theoretical framework is not provided. “A semi-structured interview guide” is not provided.
  3. How can you access the interviewees? Readers might be wondering why interviewees are willing to accept the interviews. Is there any incentive which is provided to help access the interviewees?
  4. You mention “Appendix A”. However, the appendix is not provided.
  5. It will be clearer if authors can put transcription used to support their findings in quotation marks. However, if you follow the APA or the journal guidelines, that is fine.
  6. Is it possible to use a figure showing the relationships among all stakeholders?
  7. Is it possible that a tale can be used to compare and contrast the similarities and differences between M1 and M2?
  8. In 4. Discussion, is it possible that some sub-titles can be given to help readers catch your points?
  9. In 4. Discussion, “An additional limitation to this study is the exclusive focus on the practices performed by municipal agents. It could be valuable to include input and assessments from the target group (including their parents).” The target group and their parents are crucial stakeholders. Is there any reason why they are not included?
  10. This study mentioned “consultants.” Readers, particularly from other cultures, may not understand their jobs. Are they employed by the government or not? How can they become consultants? Is there any national exam that they need to pass? “Consultants” seem to play an important role in this study. But are consultants from a public sector or from a private sector? What kind of service do consultants provide? Or is it possible that other job terms might be better than consultants?  

Reviewer 2 Report

1.1. The Danish Political and Administrative Structure

It is not clear how this chapter relates to the theoretical and practical benefits to the general public, the experiences of other countries are not reviewed.

2. Materials and Methods 162
2.1. Study Design 

Why did you choose these groups for the interview method? Maybe they are an experts? There is a lack of more detailed descriptive information about the subjects.

What practical advice would you give to people working with children with disabilities?

Reviewer 3 Report

Summary

This article addressed how multiple levels impact physical activity program for children and youth with disabilities. Specifically, these authors examined the impact of policy, organizational, and individual (municipal agents) level influences on PA implementation. The theoretical framework was a strength of the article and the ability to engage multiple stakeholders across distinct municipalities was an important methodological consideration.

General Comments

The introduction, methods, and discussion sections are very strong. There is a clear framework upon which the potential levels of influence were identified and the interview questionnaire (Appendix A) was from the associated framework literature. The connection of study findings to the framework in the discussion is also very thorough and sound.

The results section appears to be a program evaluation of two distinct municipalities rather than a collective synthesis of deductions. across multiple sites. I encourage the authors to consider revising the results section to separate research findings (RQ1-3) from the policy and municipality descriptions. The description of policies and municipalities is essential for providing context for the results; however, combining narrative results with policy/municipality descriptions makes drawing conclusions specific to the RQs difficult. Additionally, the authors reported that collecting data from multiple municipalities made the findings more robust but the analyses are mostly delimited by municipality (which limits the generalizability or external validity of findings).

Finally, the authors examined 3 important research questions but many important points in the results section addressed barriers to implementation (not specifically identified as a RQ). I recommend that the authors add a RQ on barriers to highlight these important findings or address these findings as unexpected findings of the study. Another possible option is to revise how the RQs are written to explore both facilitators and barriers of each level.

Specific Comments

Line 32  Transportation does not reflect physical activity. Did the authors intend to address mobility?

Lines 42, 45, 54  Good rationale support for the study.

Line 119   “Not least first” is a bit colloquial and may be unclear to the audience.

Line 156  How do municipalities . . . work with the implementation (policy or program implementation?)

Line 185  Remove “more on this in the results section.”

Tables 1-2  Very helpful; provide good context

Line 194  sent by

Table 3-4  If the authors revise results to analyze data collectively, it would be helpful to edit table 3 so that readers can examine similar policy topics side by side so that it is easy to determine where similarities and differences exist (the authors could still include the full table in the appendix)

Table 4  remove bullet points

Tables (general)  Left indent for the topics section may be easier to read

References

References are relevant and thorough.

Writing Clarity

The paper includes minor grammatical errors and is generally well-written.

Round 2

Reviewer 3 Report

No additional comments.